# Idiopathic Ventricular Tachycardia

**DOI:** 10.3390/jcm12030930

**Published:** 2023-01-25

**Authors:** Robert C. Ward, Martin van Zyl, Christopher V. DeSimone

**Affiliations:** Department of Cardiovascular Medicine, Mayo Clinic, Rochester, MN 55902, USA

**Keywords:** ventricular tachycardia, electrocardiogram, catheter ablation, cardiac anatomy

## Abstract

Idiopathic ventricular tachycardia (VT) is an important cause of morbidity and less commonly, mortality in patients with structurally normal hearts. Appropriate diagnosis and management are predicated on an understanding of the mechanism, relevant cardiac anatomy, and associated ECG signatures. Catheter ablation is a viable strategy to adequately treat and potentially provide a cure in patients that are intolerant to medications or when these are ineffective. In this review, we discuss special approaches and considerations for effective and safe ablation of VT arising from the right ventricular outflow tract, left ventricular outflow tract, left ventricular fascicles, papillary muscles, and moderator band.

## 1. Introduction

Ventricular tachycardia (VT) most commonly occurs in the setting of structural heart disease [1]. However, when VT is discovered in a structurally normal heart (i.e., the absence of ischemia, valvulopathies, or other cardiomyopathies), it is termed idiopathic VT. Traditionally, idiopathic VT was thought to compromise around 10% of all VTs [2]. Recently the overall incidence of idiopathic VT was found to be around 14/100,000 individuals, with similar rates between men and women [3]. In general, idiopathic VT carries a more benign prognosis relative to VT associated with structural heart disease, though it remains an important cause of morbidity and mortality in otherwise healthy individuals [4,5,6]. Idiopathic VT can often be effectively managed with medications, catheter ablation, or a combination of the two. With appropriate therapy, patients generally are not at an increased risk of sudden death and do not require implantation of an ICD.

Idiopathic VT can arise from anywhere in the heart, but certain regions are more common than others. One of the keys to accurately diagnosing VT, as well as deciding on appropriate management therapies and procedural planning, is a robust understanding of the electrocardiographic features which reflect the VT exit [7]. Multiple algorithms have been previously published to assist with localizing ventricular arrhythmias [8,9]. While referencing these algorithms to differentiate between closely adjacent structures can be useful, perhaps a more practical approach is to understand how cardiac anatomy influences the ECG signature [10,11].

Recognizing the clinical characteristics, identifying ECG signatures, knowing the relevant anatomy, and formulating a clear plan for pharmacologic or ablative intervention are key for successful management of idiopathic VT. In this review, we delineate these key attributes of the most common types of idiopathic VT (Figure 1).

## 2. Right Ventricular Outflow Tract VT

The most common location for idiopathic VT to arise from is the right ventricular outflow tract (RVOT-VT) [12,13]. Cyclic AMP mediated, calcium-dependent, triggered activity is thought to be the mechanism of this arrhythmia [14,15,16]. RVOT-VT commonly manifests in the third to fifth decades of life and is about twice as likely to occur in women compared to men [17]. In fact, triggers for RVOT-VT have some sex-specific characteristics, in particular, fluctuations in female hormones [18]. Given the mechanism, calcium loading from catecholamines and increased heart rates during exercise as well as psychological stress often precipitate the arrhythmia.

RVOT-VT is commonly found in patients with high adrenergic states (critically ill, intense exercise, etc.) As such, treadmill stress testing or prolonged ambulatory monitoring are often required to document the rhythm. Patients typically report periodic episodes of palpitations, lightheadedness, or presyncope. As stated, true idiopathic RVOT-VT is typically benign in nature [5], but it is vital to exclude arrhythmogenic right ventricular cardiomyopathy (ARVC) as a cause of the arrhythmia. In the modern era, this can be excluded by careful family history, baseline ECG, echocardiography, and consideration for cardiac magnetic resonance imaging (MRI) [19].

Given the dependency on cAMP signaling, adenosine may terminate these tachycardias [15], but beta-blockers or calcium channel blockers (CCB) are much more commonly used to prevent recurrence [20]. Antiarrhythmics including flecainide, propafenone, sotalol, or amiodarone could also be considered if beta blockers or CCB are ineffective. When medical therapies fail, consideration of catheter ablation is an appropriate alternative option. In patients who are intolerant to medications, or if medical therapy is contraindicated (such as pregnancy [21]), or in those who wish to avoid medications, catheter ablation as a first line option is also a reasonable strategy, as this can be curative.

### 2.1. ECG Signatures

The classic RVOT-VT exhibits a left-bundle pattern (QS in V1) consistent with RV activation followed by LV activation. The inferior leads (II, III, and aVF) characteristically have tall R waves, while leads aVL and aVR will exhibit deep S waves given the superior to inferior activation pattern (Figure 2). The precordial transition from negative (QS) to positive (rS/RS) in VT compared to that in sinus may help distinguish between RVOT and left ventricular outflow tract (LVOT-VT). If the precordial transition is later than that seen in sinus, the VT is likely to originate from the RVOT. Further, if the V2 transition ratio (R wave percentage of total QRS in VT divided by R wave percentage of total QRS in sinus) is <0.6, it is likely to be RVOT [22].

It is important to note that precordial lead V1 is an anterior and rightward lead and can help localize the portion of the RVOT the VT is exiting from. If a QS pattern is found in V1, this is suggestive of an anterior RVOT wall origin. If an rS pattern is seen in V1, this is suggestive of involvement of the posterior wall of the RVOT. Precordial transition is also helpful in differentiating a septal origin from a free wall origin. Transition earlier than V3 suggests a more septal location, and transition later than V3 suggests more of a free wall site [22]. Furthermore, notching of the QRS is suggestive of free wall origin compared to septum. Perhaps equally as important is close attention to lead I (a left-sided lead), which can further assist in localizing RVOT-VT. A VT exit above the pulmonary valve or in that region will originate more leftward and thus will yield a negative complex in Lead I (QS). Conversely, an origin from the rightward margin or septal RVOT would be more likely to have a positive deflection.

Important features which contribute to the ECG characteristics of RVOT-VT are the presence of supra-valvular muscle bundles, distal variable conduction (below the pulmonary valve) from extensions of muscle from these bundles, and the possibility of “dead-end tracts” or remnants of the conduction system which persist beyond the bifurcation of the bundle branches [23]. These complex interconnections can result in varied exits from a single focus and subtle beat-to-beat variation in morphology. This can be a clue to a peri-pulmonic focus.

### 2.2. Catheter Ablation

Activation mapping is the current gold standard for localizing an RVOT-VT exit [24]. However, this VT can be quite difficult to induce in the electrophysiology (EP) lab given the mechanism of this arrhythmia and the implications and limitations occurring during the procedure. To help with induction, sedation can be kept to a minimum during vascular access, and it may even be required that the patient be minimally sedated or kept awake during mapping. Pharmacologic adjuncts can be used including caffeine, isoproterenol, epinephrine, and/or phenylephrine to help with induction. Finally, both atrial and ventricular burst pacing can stimulate catecholamines and calcium loading to augment induction of VT. Pace-mapping can be used as an adjunctive strategy, especially when non-inducible, but it is important to remember a closely matched pacing morphology near the endocardial exit may be some distance from the successful ablation site. This is especially true in the case of supra-valvular or mid-myocardial/epicardial foci and is likely a common reason for unsuccessful ablation and VT recurrence.

Great care must be taken when mapping and ablating in the RVOT. Especially when the VT involves the anterior/free wall of the RVOT given the risk of perforation of the thin free wall. Our lab’s preference is to advance catheters carefully with both fluoroscopy and intra-cardiac echocardiography (ICE) guidance from the RV inflow and closely follow all the way out past the pulmonic valve. The catheter can then be safely and slowly pulled back with rotation into the septal aspect of the RVOT. The idea is to avoid pushing forward onto the free wall at any time.

ICE can be critical in identifying the anatomic location of catheters and their relation to important structures. Mapping both above and below the pulmonary valve is important and ICE can help ensure correct location of the catheters and visualize the muscle bundles above and below the valve (Figure 3). Both the thick subvalvular infundibulum and the conus papillary muscles can present challenges because contact may be difficult to maintain, thus generating trouble in achieving adequate ablation lesions. ICE can help ensure adequate contact during ablation and monitoring during energy delivery.

One should be fully aware of the anatomical relationship of the left main (LM) coronary artery and the pulmonary valve, as the LM runs along the lateral aspect of the muscular RVOT [11]. The close proximity of the target tissue to the artery may preclude ablation from the RVOT. This may require mapping and ablation from the left side of the septum if noted. In addition, when mapping the RVOT, if the earliest signals are septal, but are not very early and/or the signals are not sharp (near-field), it is imperative to map the LVOT as well. This relationship makes coronary angiography essential when ablating on the posterior RVOT to define the proximity to the artery (Figure 4).

If ablating above the PV and one is unsuccessful with a straight orientation of the ablation catheter, it may be advantageous to use a tight curve and then come down into the PV cusp (Figure 2). This can typically be accomplished by curving while in the pulmonary artery and pulling down into the cusp, which is our lab’s preference, or curving in the right ventricle and prolapsing through the PV. Regardless, cautious movements and the use of ICE are critical for safety measures (Table 1).

Outcomes for RVOT VT ablation are encouraging with acute procedural success ranging from 75 to 100% and major complications < 3% [24,25,26].

## 3. Left Ventricular Outflow Tract VT

Idiopathic VT can arise from the left ventricular outflow tract (LVOT) [27], and regions in close proximity which include the aortic valve cusps [28,29], sinus of Valsalva [30], aorto-mitral continuity, and LVOT epicardium [31]. Compared to the RVOT-VT, LVOT-VT appears to be more common in males and presents about a decade later in life [17,32]. Like with RVOT-VT, the underlying mechanism appears to be cyclic AMP mediated, calcium-dependent triggered activity [32]. Thus, it is often treated in a similar fashion with conservative medical management, beta-blockers or CCB, consideration of anti-arrhythmic drugs, and/or catheter ablation.

### 3.1. ECG Signatures

Representative ECGs for the LVOT-VT can have wide variation given the complexities noted at different parts of this region of the heart. Regardless, the ECG still can provide valuable insight on discerning origins including the aortic sinuses of Valsalva (right, left, and non-coronary), aorto-mitral continuity, and LVOT epicardium. VT arising from the sinuses of Valsalva typically exhibit a left-bundle pattern with a strong inferior axis (i.e., positive R waves in leads II, III, and aVF). In lead V1 and V2, an R wave duration of greater than 50% of the entire QRS duration and/or an R/S amplitude ratio of >30% is highly suggestive of aortic sinus of Valsalva origin and can help differentiate from RVOT [29]. VT from the right coronary sinus of Valsalva (RSoV) will exhibit an early precordial transition in V2 or V3 and typically has a tall R wave in lead I (right to left). VT from the left coronary sinus of Valsalva (LSoV) classically has a negative QRS in Lead I (left to right) with a very early precordial transition around V2 (Figure 5). As previously discussed, when the precordial transition occurs at lead V3, comparing the transition in VT to the transition in sinus can be helpful in distinguishing RVOT from LVOT-VT. In particular, if the VT transition is earlier than sinus and the transition ratio is >0.6, this is suggestive of LVOT-VT [22]. These ECG findings are helpful in procedural planning but in no way are 100% predictive. It is common to require mapping in both the RVOT and LVOT to find the earliest and safest location to ablate (Figure 6).

VT arising from the aorto-mitral continuity (AMC) has a characteristic ECG pattern with a right bundle configuration (positive QRS) in V1, inferior axis, positive QRS in lead I, and positive concordance across the precordial leads with no S wave in V6. A specific but insensitive ECG characteristic of AMC-VT is a qR pattern in V1, found in about 25% of cases [33].

An LVOT epicardial origin characteristically has a left-bundle morphology, inferior and rightward axis. The R wave amplitude in the inferior leads and Q wave amplitude in aVR and aVL are very prominent (with aVL usually being more negative than aVR). V1 can be variable in morphology, and precordial transition can occur as early as V2 [31]. An R wave pattern break at lead V2, i.e., loss of or reduction in R wave amplitude at lead V2 compared to leads V1–3, is also associated with an epicardial origin [34].

### 3.2. Catheter Ablation of LVOT-VT

Similar to RVOT-VT, the LVOT-VT can be difficult to induce due to the underlying mechanism. Often, strategies discussed in the RVOT-VT section can be of similar use in these cases, i.e., reducing or avoiding sedation, utilizing pharmacologic adjuncts for stimulation, and attempting rapid burst pacing.

A retrograde approach is utilized to map above and in the depths of the sinus of Valsalva, as well as below the cusps (Figure 7). This also allows for access to the peri-aortic LV endocardium. A trans-septal approach is feasible and may offer a safety benefit by avoiding risk of injury to the aorta, embolization of aortic atheroma, coronary dissection, or valvular injury, or avoidance if aortic valve replacement was performed (Figure 8). However, this approach requires more complex catheter movements to properly get below the valve and to achieve adequate stability. In contrast, using a retrograde access approach via the aorta and into the LV, is simpler, as this involves pulling the catheter back to the valve with the same flexion used to safely cross the valve.

Anatomically, the LV summit is delineated as a triangle between the left anterior descending and left circumflex arteries; it is divided by the great cardiac vein/anterior interventricular vein junction into an inaccessible region superiorly and an accessible region inferior and laterally [35]. Ablating VT at the LV summit region can be challenging due to the thick muscle, multiple converging anatomic relationships (left sinus of Valsalva above and below the valve, RVOT, and GCV/AIV junction branches of the coronary sinus), and proximity to the epicardial coronary arteries. These cases require extensive forethought and planning. Our lab will often place a catheter in the coronary sinus and advance deep into the GCV/AIV region (Figure 7) to adequately bracket the VT location, as well as to provide anatomic reference on fluoroscopy and the mapping system. Very often the tissue is mid-myocardial or even closer to the epicardial surface and thus ablation from the endocardium vantage points (above and below the left and right sinus of Valsalva), higher power and longer duration (several minutes) are usually required. One should consider mapping and ablating adjacent chambers for an anatomical vantage point even if slightly later or far-field as long as it is safe.

Ablation within the distal coronary sinus and its distal branches has the issue of high impedance and coronary artery proximity which limits the ability to deliver adequate power, if at all, and may preclude ablation. One should always perform a coronary angiogram to delineate coronary artery anatomy and proximity prior to ablation from this region. Increasing irrigation flow prior to coming on ablation and gradual uptitration from a lower power can be helpful given the high impedance in this region. Subxiphoid epicardial access has at times been utilized but is usually not successful given the location relative to the coronary artery which precludes ablation, as well as the epicardial fat that precludes adequate ablation energy delivery to the key region [36]. Challenging cases may require experimental techniques such as coronary venous branch or septal arterial ethanol ablation [37,38], surgical access for direct coronary visualization and separation from the myocardial site of interest [39], or direct intramyocardial ethanol injection [40] (Table 2).

Similar to RVOT VT, ablation outcomes of LVOT VT are overall good, but vary depending on the location [25,26,34,41].

## 4. Idiopathic Left Ventricular Tachycardia (Fascicular VT)

Idiopathic left ventricular tachycardia (ILVT) is also known as fascicular or “verapamil-responsive” VT. The characteristics were first described in the late 1970’s and early 1980’s [42,43]. Patients often present in the 3rd to 5th decade of life with repeated episodes of palpitations or presyncope, and it is more commonly found in males [44,45]. The tachycardia can occur at rest but is quite often precipitated by exercise or increased sympathetic tone. True syncope is rare, and overall prognosis is very good. Originally the mechanism was postulated to be triggered activity, but contemporary studies and descriptions strongly support a re-entry mediated mechanism [46,47,48,49], involving a portion of the left fascicular system and a localized region of slowly conducting myocardium.

Given the unique sensitivity, verapamil is the first line treatment of choice in the acute setting, assuming hemodynamic stability and the correct diagnosis has been made. Careful consideration of other diagnoses should be given prior to administering verapamil if the diagnosis is ambiguous. Caution is especially needed if the patient is hypotensive, given the vasodilation that occurs with verapamil and concurrent reduction in cardiac output. Once a patient has been established as having ILVT, if symptoms recur, a trial of scheduled verapamil can be considered. However, if tachycardia recurs, typically the patient should be referred for ablation, as success rates are high.

False tendons (FT) can be implicated in fascicular VT and are important to recognize. The prevalence of left ventricular FTs exceeded 50% in one autopsy study [50]—though most of these will not be involved in ventricular arrhythmogenesis. Series so far have shown a wide age range at presentation and no sex predominance [51,52]. The mechanism by which FT result in VT remains speculative but could include reentry utilizing these structures (which often contain and are closely associated with conduction tissue), triggered activity, abnormal automaticity, or mechanical stretch related ectopy. The close interpolation of FTs with conduction system tissue raises potential for ectopy to trigger ventricular fibrillation [53].

### 4.1. ECG Signatures

Idiopathic Fascicular VT, because of its involvement with the native conduction system, can be difficult to discern from SVT as it appears narrower compared to the typical wide-complex VT. The two most common subtypes include left posterior and left anterior fascicular origins, with the left posterior fascicle being most common. Both exhibit a rather typical RBBB morphology with a sharp upsloping and narrowish QRS (<120–140 ms). Posterior fascicular VT exhibits a left superior axis with RS complexes in V5–6 (right bundle, left anterior fascicular block pattern). Left anterior fascicular VT exhibits a right or right inferior axis (right bundle, left posterior fascicular block pattern). Therefore, left posterior fascicular VT will typically exhibit a predominately positive lead I, while anterior fascicular VT will exhibit a negative or neutral lead I. Differentiating from SVT non-invasively requires the presence of reproducible AV dissociation or fusion/capture beats on ECG (Figure 9). In a patient with a structurally normal heart and normal infrahisian conduction on ECG during sinus rhythm, development of a bifascicular block pattern during conducted SVT would be highly unusual and should raise a strong suspicion for fascicular VT.

The ECG characteristics of VT related to anatomic variants including FTs are highly varied but can often be deduced from the location of these structures within the heart. Given the close relationship of left ventricular FTs found inserting on the left ventricular septum with the specialized conduction system, the ECG features of VT with a septal exit may be indistinguishable from those of fascicular VT.

### 4.2. Catheter Ablation

IL-VT is notoriously difficult to induce, with one series suggesting up to 40% of patients are non-inducible [54], which can be a limitation to ablation. Atropine, isoproterenol, and phenylephrine are all pharmacologic adjuncts that can be helpful. Certain pacing maneuvers including burst pacing, triple ventricular extra stimuli, and short-long-short sequences have been shown to be helpful [55]. Conventional mapping, pace mapping, and entrainment techniques are challenging given the complexity of the fascicular system and difficulty in capturing discrete fibers without capturing underlying muscle.

Ablation should ideally target the retrograde limb of the conduction tissue proximal to the turnaround point of the theoretical reentry circuit, while preserving the more proximal conduction system to avoid conduction system damage (Figure 10). A comparative mapping approach using the HV interval in sinus rhythm versus VT can provide a level proximal to the turnaround site. The equation proposed is: [(HV_NSR_ + HV_LFTA_)/2] [56]. An approach for mapping P1/P2 using linear multielectrode catheters is interesting from a mechanistic perspective [57]. P1 represents a critical zone of slow conduction that manifests as a diastolic potential during VT that is linked to P2, which represents the distal left posterior fascicle (Figure 11). If P1 can be identified, an ablated termination of the VT is often successful. However, the technique is challenging to perform and requires meticulous multielectrode catheter placement, close attention to signals and proper signal annotation. Another option involves transection of the distal left posterior fascicle to eliminate the ability of the tachycardia to continue. This is easier to accomplish and is practical when VT cannot be induced. It is expected that this approach will be left with a corresponding fascicular block post-ablation and indeed this is one way to monitor during ablation. ICE is crucial to look for anatomic variants, such as false tendons, which may also be targets for mapping and ablation of critical components to the circuitry (Figure 12).

**Figure 10 jcm-12-00930-f010:**
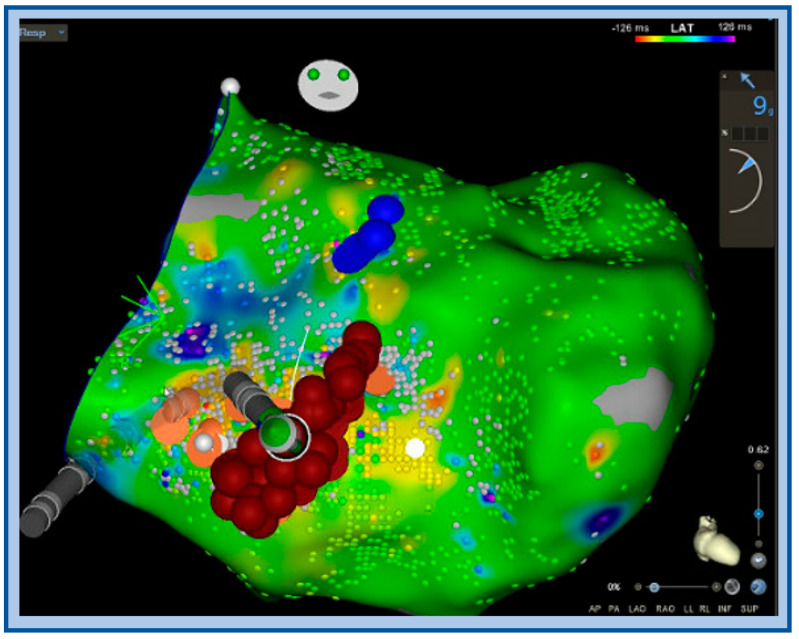
Activation map with ablation lesions transecting the left posterior fascicle (orange dots) blue dots denote the left anterior fascicle, ablation was performed to transect the LPF at this level, which also included very fractionated signals locally.

**Figure 11 jcm-12-00930-f011:**
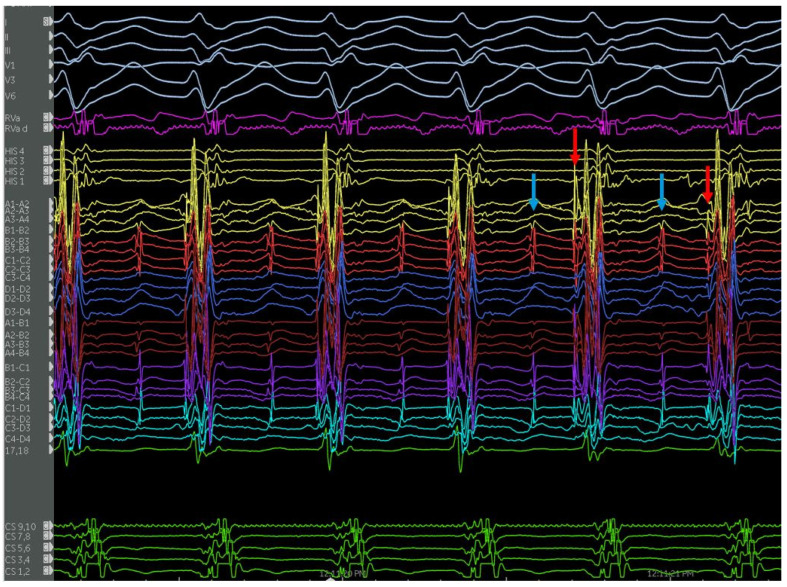
Endocardial signals of IL-VT showing typical pre-systolic signals. Blue arrows denote the P1 signal and red arrows denote the P2 signal during tachycardia.

**Figure 12 jcm-12-00930-f012:**
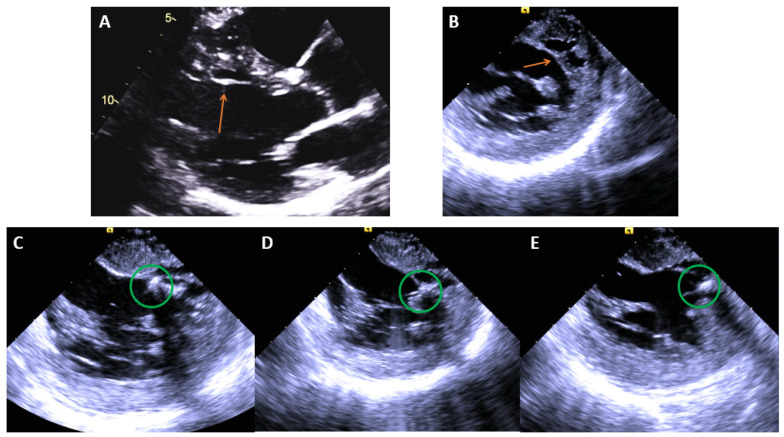
(**A**) Echo image showing a false tendon (orange arrow) on the anteroseptum of the LV, (**B**) ICE image showing the same structure. (**C**–**E**), ICE images of cryoablation on the false tendon with ablation lesions spanning the proximal, mid, and distal segments moving left to right noted by the green circles (Table 3).

During ablation there are several issues that can arise, and the operator should be aware of these and be prepared to act to ensure patient safety. Radiofrequency energy can trigger complex fascicular ectopy and even induce VF, which could also become incessant. This usually requires continued ablation to eliminate the VF. The operator should be prepared with IV lidocaine, IV amiodarone, and the capacity to quickly defibrillate should be available prior to coming on ablation. Furthermore, alerting the ECMO team prior to ablation is of great importance in the case that incessant VT/VF occurs.

## 5. Papillary Muscle VT

Papillary muscles are closely associated with several intracavitary structures, and these pose unique challenges for mapping and ablating this type of VT. Anatomical changes such as abnormal papillary muscles with multiple heads and attachments to the mitral valve, calcifications, contact lesions, and associated structures such as trabeculations and false tendons are important potential substrates for idiopathic VT in what could otherwise be considered a structurally normal heart [52,58].

Much more common is locating the origin on the papillary muscles themselves, whether this involves the tip, base, in between two muscle heads, or at the insertion sites. All are within the range of variability and need to be carefully interrogated with echocardiography or cardiac MRI. Specific findings to evaluate are contact lesions from the mitral valve apparatus as well as calcifications or echodense areas noted on imaging. Carefully interrogating the ventricles with imaging for these subtle anatomic variants is key for establishing a potential etiology and ablation target, which may otherwise be missed by cardiologists and radiologists interested in excluding gross structural disease rather than arrhythmic substrate. ICE dramatically improves yield and can be instrumental for successful ablation.

### 5.1. ECG Signatures

LV papillary muscle VT will exhibit a RBBB pattern and a broader QRS (>130 ms) [59]. The anterolateral papillary muscle VT exhibits a right inferior axis but can display inferior lead discordance with a negative lead II and positive lead III. Lead I will be predominantly negative with an rS pattern. These typically have a precordial transition to R/S configuration from V4–V6. V6 will often have an R/S ratio < 1 [60]. A posteromedial papillary muscle VT exhibits left superior axis [61]. V1 will typically exhibit a qR or R pattern. Lead I is usually positive with an R wave. Precordial transition can be variable but commonly is later, again with an R/S configuration near V4–V6 (Figure 13).

### 5.2. Catheter Ablation Considerations

While commonly successful, ablation of these sites is often challenging. Pace mapping is almost of no utility given all the surrounding structures/tissues that can be captured with pacing output, catheter ectopy, and catheter stability [61]. The VT may also exhibit pleomorphic morphologies, potentially due to multiple exit sites [62]. Once mapped, it is difficult to maintain ablation catheter contact. ICE can be very useful in fully realizing the relevant anatomy and confirming contact on the appropriate structure. (Figure 14). If contact continues to be an issue, cryoablation can help. Ablating a wide area around the base of the papillary muscle is commonly required to fully suppress the VT [52], although electrically isolating the papillary muscle may have detrimental hemodynamic effects (Table 4).

Acute success of papillary muscle VT ablation has been reported slightly lower than outflow tract VT in some series [25] but has been reported as high as 100% in others [62].

## 6. Moderator Band VT

The moderator band (MB) is a muscular structure that typically runs from the RV anterior septal wall to the lateral RV papillary muscle, varies in morphology, and is found in over 90% of human hearts [63]. It typically houses Purkinje cells involved in a distal extension of the right bundle branch, allowing for rapid conduction to the lateral wall. The structure has increasingly been recognized as a nidus for a variety of arrhythmias, including idiopathic VT [64].

The underlying mechanism is not fully elucidated but multiple theories have been proposed: afterdepolarizations related to Purkinje cells [65], mechanical stretch induced electrical changes precipitating ectopy [66], and possibly substrate for a macro-reentrant circuit as supported by experimental evidence [67]. The main cause for concern with MB-VT is that it has potential to degenerate into VF, and can also present with polymorphic VT. Because of those associated risks, aggressive treatment often with catheter ablation is typically recommended. Ablation can be highly successful in suppression of the arrhythmia [64].

### 6.1. ECG Signatures

MB-VT exits near the apicolateral RV and therefore exhibits a LBBB morphology. The QRS morphology can have subtle variation which can be a tip-off due to varying exits and fusion. The axis is leftwards and superior with predominantly positive QRS in lead I and negative QRS in the inferior leads (Figure 15). Given early involvement of the specialized conduction system, the QRS may be relatively narrow with a sharp upstroke. However, this is dependent on the exit site and could be wide if the exit is closer to the RV free wall. Further, the precordial transition is typically later than V4 and later than that in sinus [64,68]. A common feature is discordance between the inferior leads with a dominant positive deflection in lead II and a negative deflection in lead III, as the VT is exiting closer to lead III [68]. Like idiopathic ILVT, MB-VT may be difficult to distinguish from SVT with left branch block aberrancy by ECG alone in the absence of AV dissociation or fusion/capture beats. Again, development of rate dependent LBBB (as opposed to isolated RBBB) aberrancy with tachycardia in a structurally normal heart is unusual and should raise suspicion for MB-VT. As an aside, the differential diagnosis, especially in younger patients, includes atriofascicular tachycardia.

### 6.2. Catheter Ablation

Conventional mapping techniques are typically effective. However, they are complex because the moderator band is large, can have variable septal and lateral insertions that may need to be targeted, and pose difficulty in catheter stability. Both the septal and lateral insertions of the MB should be mapped, especially if there are two variable exits noted during EP study or from ECG and Holter monitoring. Pacemapping can be helpful, but it requires care to confirm which structures are being captured. ICE is critically helpful to both define the MB structure and ensure appropriate mapping and ablation catheter contact. Cryoablation can help improve contact with the MB to allow for ablation on the body of the structure as well as both insertion sites (Figure 16). As with ablation of ILVT, radiofrequency energy on the moderator band can induce VF. IV amiodarone, IV lidocaine, and the capacity to quickly defibrillate should be available prior to ablation and patients should be counseled about the possibility (Table 5).

## 7. Summary

Idiopathic ventricular tachycardia, defined by the absence of structural heart disease, can arise from anywhere within the heart, but commonly occurs in select locations. Many of these arrhythmias carry more benign prognoses than VT associated with ischemia or structural heart disease, but there are important characteristics to consider. A robust understanding of the relevant anatomy, underlying pathophysiology, and ECG signatures is crucial to effective management, as we have outlined. Treatment regimens often include medications such as beta blockers, calcium channel blockers, or antiarrhythmics. Most of the various subtypes can also be successfully treated with catheter ablation. It is important to have a deep understanding of the surrounding anatomy in order to aid in mapping and safe and effective ablation.

## Figures and Tables

**Figure 1 jcm-12-00930-f001:**
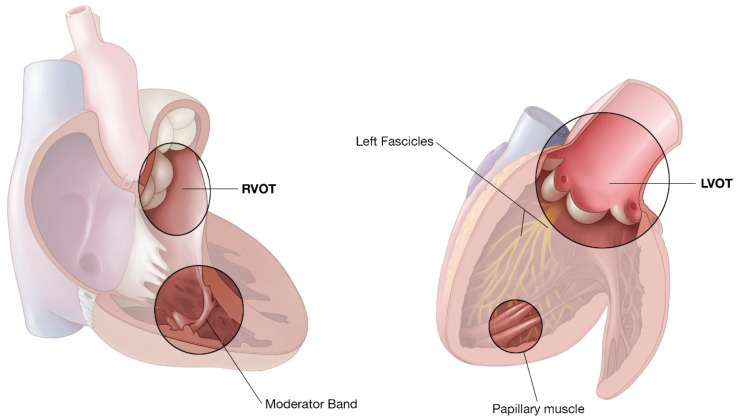
Representative illustration of the right ventricle and left ventricle highlighting common locations of idiopathic VT.

**Figure 2 jcm-12-00930-f002:**
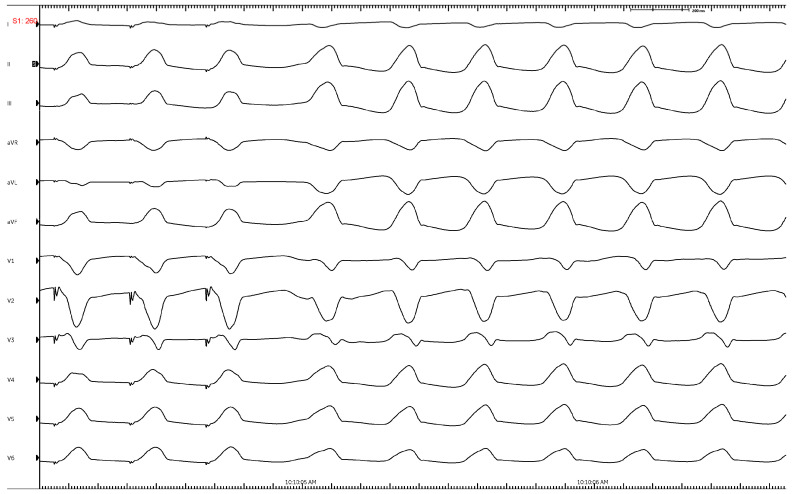
Characteristic ECG of an RVOT-VT showing LBBB, inferior axis, transition at V3, lead 1 is negative. Successful ablation was performed just below the pulmonary valve on the leftward free wall.

**Figure 3 jcm-12-00930-f003:**
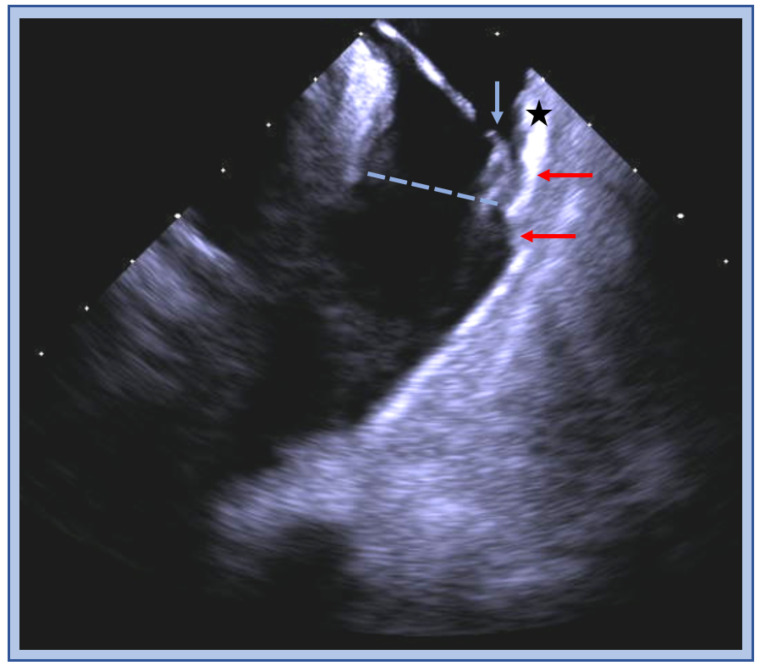
ICE image of the RVOT with an ablation catheter (blue arrow) positioned on the anterior wall (star) below the pulmonic valve plane (blue dashed line), thick muscle bundles (red arrow) are visualized above and below the valve.

**Figure 4 jcm-12-00930-f004:**
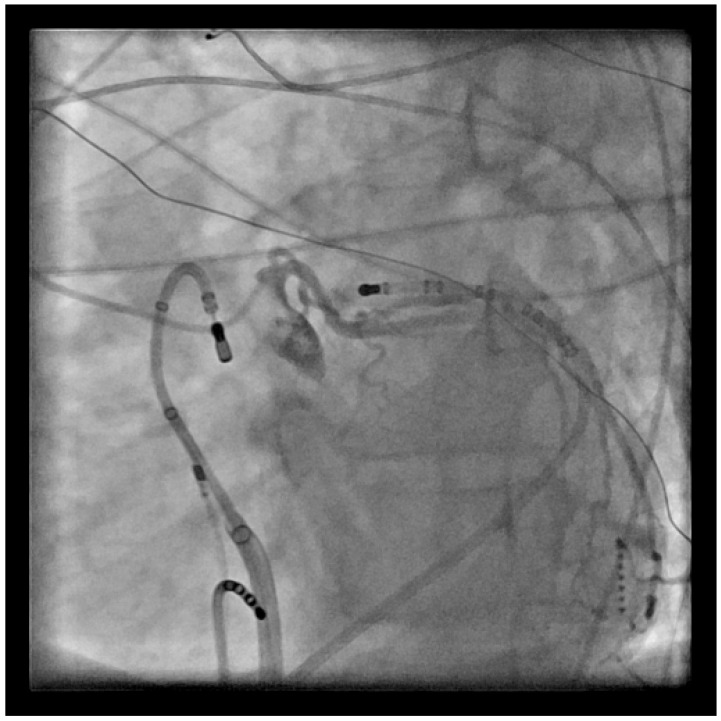
LAO view with ablation catheter in the RVOT retroflexed back onto the PV, and a left coronary angiogram showing the proximity to the catheter location.

**Figure 5 jcm-12-00930-f005:**
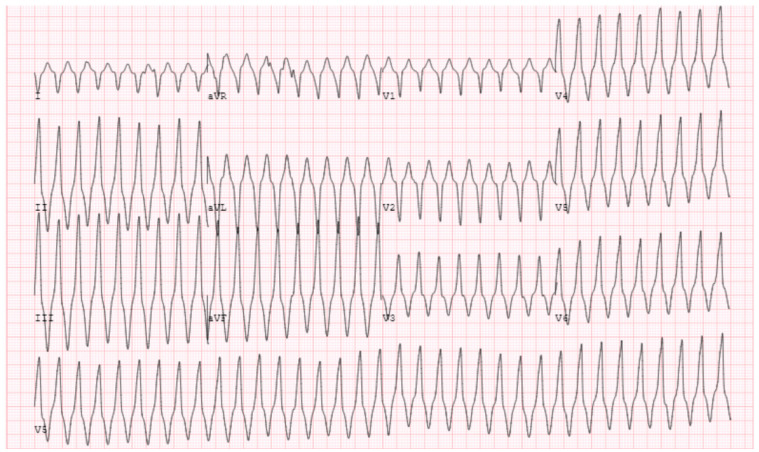
ECG from LVOT showing a rightward inferior axis with early transition at V3. Successful ablation was performed at LSoV. This also required ablation at the AMC region.

**Figure 6 jcm-12-00930-f006:**
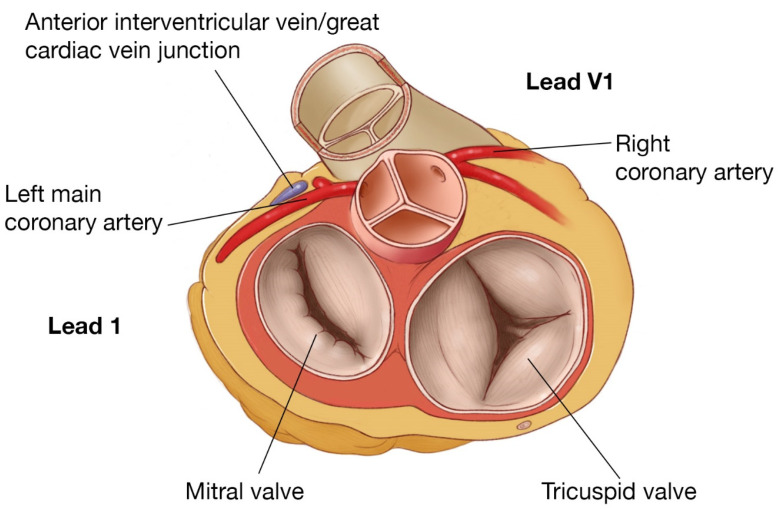
RVOT/LVOT anatomy, showing proximity of posterior RVOT to the left main and anterior LVOT and the relationship of leads I and V1.

**Figure 7 jcm-12-00930-f007:**
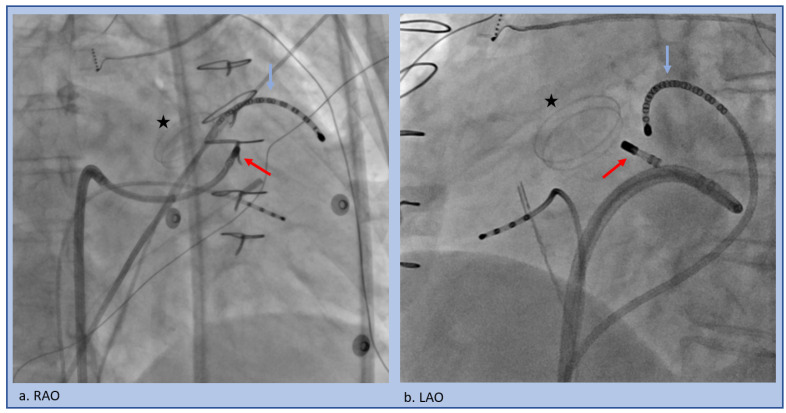
Trans-septal approach to the LSoV in the setting of a mechanical aortic valve (star). A multi-electrode catheter (blue arrow) is pushed through the coronary sinus into the Great Cardiac Vein/Anterior Interventricular Vein junction to span the relevant area. The ablation catheter (red arrow) traverses the trans-septal access and then is flexed back to underneath the LSoV (**a**) RAO projection, (**b**) LAO projection.

**Figure 8 jcm-12-00930-f008:**
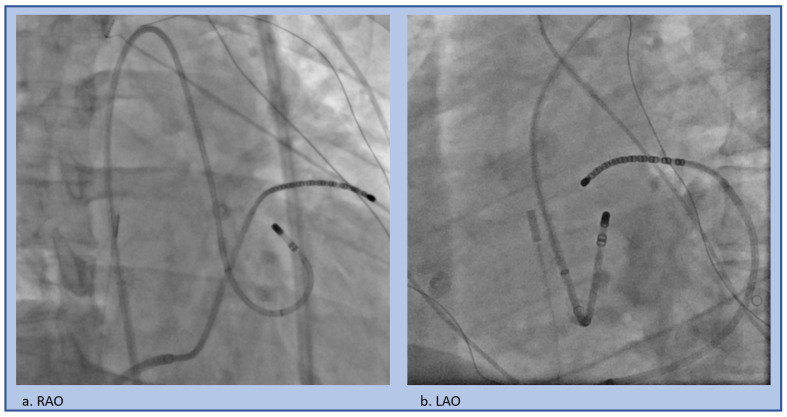
Retrograde approach to the LSoV, (**a**) RAO projection, (**b**) LAO projection.

**Figure 9 jcm-12-00930-f009:**
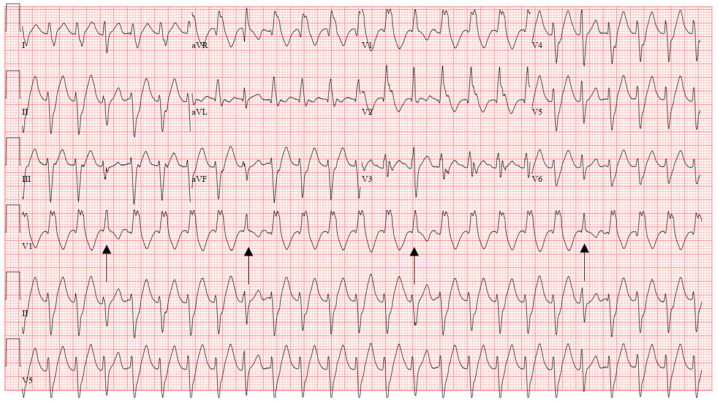
Characteristic ECG of fascicular VT showing RBBB and LAFB pattern, diagnosis is clinched by the fusion beats (arrows). Successful ablation was performed by transecting the left posterior fascicle.

**Figure 13 jcm-12-00930-f013:**
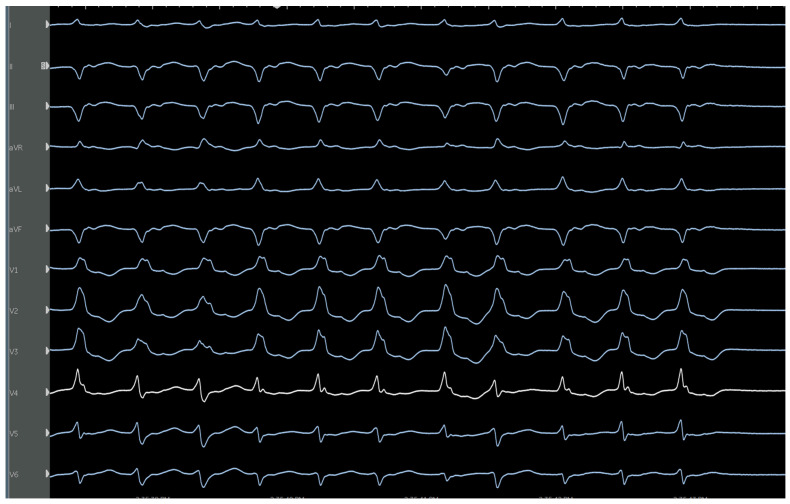
Characteristic ECG of posteromedial papillary muscle VT with a RBBB, left superior axis, and later precordial transition to R/S configuration.

**Figure 14 jcm-12-00930-f014:**
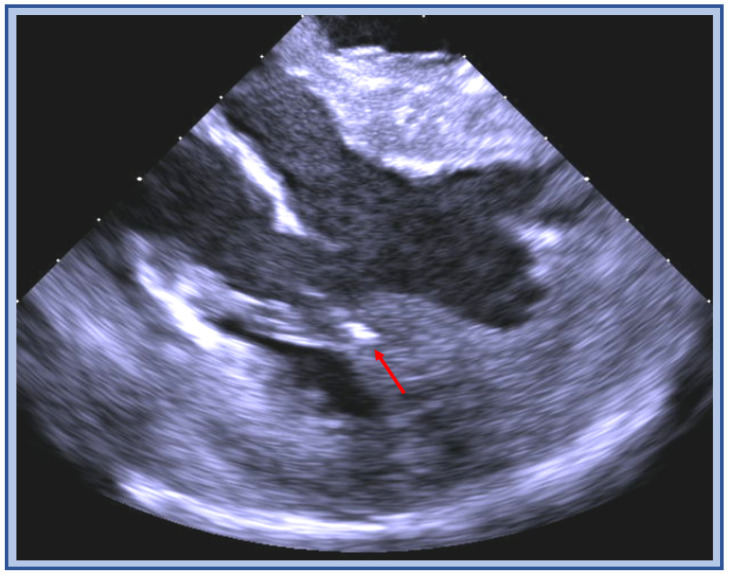
ICE image of cryoablation catheter positioned in between the two heads of the papillary muscle (red arrow) during ablation for added stability.

**Figure 15 jcm-12-00930-f015:**
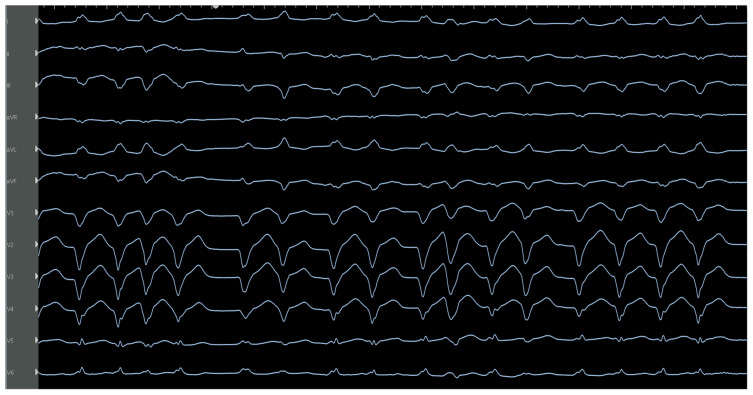
Characteristic ECG of moderator band VT-showing LBBB, left superior axis with S wave more negative in III than II, transition occurs late at V5 (given the apical exit) and variable QRS morphology is seen. Successful ablation was performed on the moderator band towards the insertion onto the papillary muscle close to the free wall.

**Figure 16 jcm-12-00930-f016:**
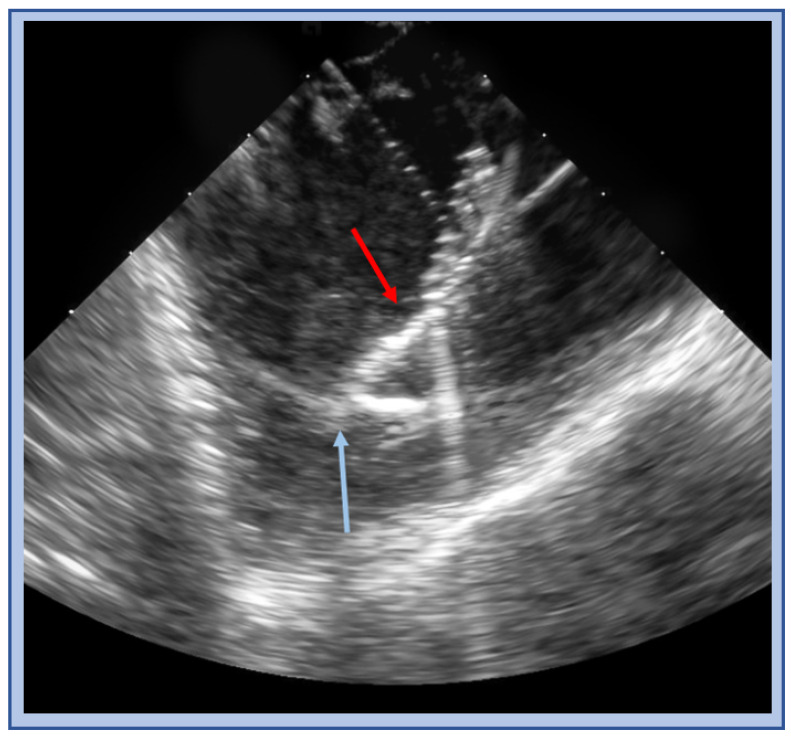
ICE image of a cryo catheter (red arrow) balancing on and adherent to moderator band (blue arrow).

**Table 1 jcm-12-00930-t001:** Summary of RVOT ablation tips.

Right Ventricular Outflow Tract Ablation Tips
Minimize sedation and consider pharmacologic adjuncts to induce VT
ICE is critical for safe navigation in the RVOT, contact/lesion formation, and localization of structures such as pulmonary valve and coronary arteries
Posterior RVOT ablation requires a coronary angiogram to safely define proximity
Great care must be taken with catheter manipulation, in addition to utilization of ICE and fluoroscopic guidance, to stay septal/posterior to go beyond the pulmonic valve

**Table 2 jcm-12-00930-t002:** Summary of LVOT ablation tips.

Left Ventricular Outflow Tract Ablation Tips
Retrograde approach is often preferred, especially with ablation above valve or in depth of the Sinus of Valsalva
Transseptal is useful, especial with severe coronary disease, aortic atheroma, or a mechanical aortic valve
ICE is critical for safe catheter movements in the aorta and Sinus of Valsalva
A multipolar CS catheter should be used and advanced far out to span the GCV/AIV region, in order to bracket the earliest ventricular electrogram. This can be used as a fluoroscopic and electroanatomic mapping target for mid/epicardial location
Use caution with ablation in the distal CS/GCV/AIV given proximity to arteries and high impedance, Coronary angiography must be performed

**Table 3 jcm-12-00930-t003:** Summary of IL-VT ablation tips.

Idiopathic Left Ventricular VT Ablation Tips
Atropine, Isoproterenol, and Phenylephrine are pharmacological agents useful for induction n
Burst pacing, triple stimuli, and long-short sequences can help induce
ICE is critical to look for anatomic variants (false tendons)
When the VT cannot be induced, transection of the distal left posterior fascicle is a reasonable approach
RF energy can trigger complex ectopy and even induce VF, which may become incessant

**Table 4 jcm-12-00930-t004:** Summary of papillary muscle ablation tips.

Papillary Muscle VT Ablation Tips
Pace-mapping is of limited utility given the proximity to multiple tissues in this complex region and variable exits
ICE is critical to visualize the papillary muscles and to confirm contact during mapping and ablation
Cryoablation is particularly helpful to achieve stability at the tip of the papillary muscle
Combination of radiofrequency to encircle the papillary muscle and ablation at the base can be helpful and sometimes is necessary for success

**Table 5 jcm-12-00930-t005:** Summary of moderator band ablation tips.

Moderator Band VT Ablation Tips
Both the septal and lateral insertions of the moderator band should be mapped
Pace mapping can be helpful, but requires careful attention to confirm which structures are being captured
ICE is helpful to define the moderator band structure and ensure appropriate contact
RF energy on the moderator band can induce VF

## Data Availability

Not applicable.

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
