# Peer review of "Idiopathic Ventricular Tachycardia"

_jcm, 2023, doi:10.3390/jcm12030930_

Round 1
Reviewer 1 Report
In this manuscript, Ward et al. reviewed the mechanisms, clinical/ECG presentations and medical and ablation strategies of “Idiopathic ventricular tachycardia” including RVOT, LVOT, fascicular, Papillary muscle and moderator band VTs.
The review is overall well written.
I have some comments=
1- This topic, also known as VT in structurally normal hearts, was a subject for multiple reviews (JACC VOL. 70, NO. 23, 2017). I don’t see what’s really new in the current review. It would be interesting to emphasize new concepts included in the manuscript.
2- I find the figures not nicely presented and many figures (mainly ICE figures) are of poor quality. Usually in such long reviews, figures are very important and should include the maximum of information.
a. Please include 12-lead ECG with classical amplifications (25mm/s-10mm/mV)
b. improve the quality of ICE images by manually reconstructing and annotating each structure,
c. Some figures require corrections=
Central figure= LVOT is below the aortic valve
Arrows in figure 10 are not in the right place
3- It is very surprising to the reviewer that no endocardial electrogram has been shown in the manuscript. The reviewer understands that anatomical approach is becoming more commonly used, however, ablation is still based on electrograms in this kind of arrhythmia
4- The authors emphasized the role of ICE in guiding ablation. I think it should be fair to discuss other strategies like using CT scans with image integration, the use of contact force…
5- An important section is missing which is the outcome after catheter ablation. Please include these data for each subtype of arrhythmia.
Minor comments=
- Line 68= Are you sure that pregnancy is a good indication for ablation? If yes, please add a reference.
- Line 71= QS pattern in V1 is not always consistent with RV activation followed by LV activation as it may be seen in septal arrythmias
- Line 275= “Pre-irrigation prior to coming on ablation, gradual uptitration from a lower power, and the use of hypertonic saline can be helpful given the high impedance at this region” = I did not understand well the concept of Pre-irrigation, did the author mean increasing the irrigation flow when ablating inside the CS like we usually do? I think Hypertonic saline is not commonly used, please add a reference to show safety of such approach
- A classical approach in inducing Purkinje arrhythmias is pacing the atria, it may be interesting to add it in corresponding sections
Reviewer 2 Report
It is my pleasure to review this review paper about idiopathic VT. The paper is well written and deserves publication. I have some suggestions for authors.
#1. This review paper does not include sample electrocardiograms. Please provide
- Example of P1 (especially during ongoing VT) and P2 potentials
- Appropriate potentials to deliver RF energy in RVOT and LVOT VT. Sample unipolar potentials can also be helpful.
#2. Please provide suggestions to eliminate PVCs arising from deep musculature or somewhere between RVOT and LVOT?
#3. Can authors discuss about preferential conduction of PVCs? Some PVC have changing exit site during ablation making successful ablation difficult.
#4. Ablation of papillary muscle often requires massive delivery of RF energy. Sometimes encirclement is required as the authors mentioned. However, not much is known about the safety of such extensive ablation of papillary muscle. Can authors provide expert opinion about safety of papillary muscle ablation?
#5. Cardiac perforation and consequent cardiac tamponade is a major concern. Please provide adequate management of cardiac tamponade during VT ablation. Management strategy might differ between RV and LV perforation.
#6. Can authors describe at least in brief about potential complications and their management strategy during idiopathic VT ablation?
#7. Any suggestions for deep-seated VT locations? Such as needle ablation or pulsed-field ablation?
Round 2
Reviewer 1 Report
The authors answered my questions, I don't have further comments.